# Effects of Andrographolide-Loaded Nanostructured Lipid Carriers on Growth, Feed Efficiency, and Resistance to *Streptococcus agalactiae* in Nile Tilapia (*Oreochromis niloticus*)

**DOI:** 10.3390/ani15142117

**Published:** 2025-07-17

**Authors:** Warut Kengkittipat, Manoj Tukaram Kamble, Sirikorn Kitiyodom, Jakarwan Yostawonkul, Gotchagorn Sawatphakdee, Kim D. Thompson, Seema Vijay Medhe, Nopadon Pirarat

**Affiliations:** 1International Graduate Course of Veterinary Science and Technology (VST), Faculty of Veterinary Science, Chulalongkorn University, Bangkok 10330, Thailand; warut_ceo@warutosama.com; 2Center of Excellence in Wildlife, Exotic, and Aquatic Animal Pathology, Faculty of Veterinary Science, Chulalongkorn University, Bangkok 10330, Thailand; taregust@hotmail.com (S.K.); gotchagorn.sd@gmail.com (G.S.); 3National Nanotechnology Center (NANOTEC), National Science and Technology Development Agency (NSTDA), Pathumthani 12120, Thailand; jo_jak49@hotmail.com; 4Moredun Research Institute, Penicuik EH26 0PZ, UK; kim.thompson@moredun.ac.uk; 5ASEAN Institute for Health Development, Mahidol University, Salaya, Putthamonthon, Nakhon Pathom 73170, Thailand; seemamedhe@gmail.com

**Keywords:** Nile tilapia, andrographolide, nanostructured lipid carriers (NLCs), *Streptococcus agalactiae*, aquaculture, antibacterial activity, growth performance, disease resistance, bioavailability

## Abstract

Tilapia farming plays a vital role in providing affordable, nutritious food worldwide. However, disease outbreaks—particularly those caused by harmful bacteria such as *Streptococcus agalactiae*—can lead to high fish mortality and severe economic losses. To reduce reliance on antibiotics, which contribute to resistance and environmental concerns, researchers are exploring safer, plant-based alternatives. In this study, we used a natural compound, andrographolide, derived from *Andrographis paniculata*, and enhanced its efficacy by incorporating it into a lipid-based nanocarrier system. This formulation protected the compound and improved its absorption in the fish. When added to the tilapia diet, the new formulation significantly boosted growth, improved feed utilization, supported liver health, and most importantly, enhanced resistance to bacterial infection. Compared to conventional treatments or the plant compound alone, the nanostructured lipid carrier system delivered markedly better outcomes. These findings suggest that integrating natural bioactives with advanced delivery technologies can help farmers raise healthier fish more sustainably. This approach could reduce chemical use in aquaculture, support environmental protection, and improve global food security by making fish farming more productive and resilient.

## 1. Introduction

Aquaculture has emerged as one of the fastest-growing sectors in agriculture, driven by the rising global demand for fish protein in response to population growth [1,2]. Among freshwater farmed species, Nile tilapia (*Oreochromis niloticus*) ranks second globally, with a production volume reaching 5.3 million tonnes in 2022 [3]. This species is widely cultivated due to its strong market demand, rapid growth, high economic returns, and tolerance to varying environmental conditions [4]. To increase production, farmers often adopt intensive aquaculture systems; however, intensification can compromise sustainability by increasing the risk of disease outbreaks and resulting in significant economic losses [5,6]. Nile tilapia is particularly vulnerable to streptococcosis, a bacterial disease not always named for the genus but with considerable impact on tilapia farming [7,8].

*Streptococcus agalactiae* (Group B Streptococcus; GBS) is a major pathogen in commercial aquaculture worldwide and is responsible for severe outbreaks in farmed fish [9]. In Thailand, repeated incidents of high mortality due to GBS have been reported in Nile tilapia reared in cage and earthen pond systems [10,11]. Among the genotypes isolated, the β-hemolytic serotype Ia strain is most commonly detected in Thai aquaculture [10]. Genomic analysis has revealed that these isolates belong to sequence type (ST) 7 within clonal complex (CC) 7 and include strains such as FNA07, FPrA02, and ENC06—genotypes known to cause septicemia in fish and opportunistic infections in humans [12,13]. Although these strains share serotype Ia and sequence type ST7, they exhibit minor genetic variation in virulence loci such as *scpB* and *cylE*, which may influence their pathogenicity [14,15]. Experimental infections in Nile tilapia have demonstrated variable clinical outcomes among strains, supporting the notion that isolates like ENC06 may cause more severe systemic disease [16,17].

In commercial tilapia farming, antibiotics and chemotherapeutic agents have traditionally been used to control bacterial diseases [18]. However, the overuse and misuse of these compounds have raised concerns about the emergence of antibiotic-resistant bacteria, environmental pollution, and consumer safety issues [19]. Consequently, interest has grown in alternative approaches, including plant-based extracts [8,20,21,22], probiotics, and prebiotics [23] for disease control in aquaculture.

Among various plant-derived compounds, andrographolide (AND)—a diterpenoid lactone from *Andrographis paniculata*—has shown considerable potential due to its wide-ranging pharmacological activities, including antimicrobial, antifungal, anticancer, hepatoprotective, anti-inflammatory, and immunomodulatory effects [24,25,26,27]. Although AND has been more widely studied for its antiviral and anti-inflammatory properties, emerging evidence suggests that it also possesses antibacterial activity, particularly against Gram-positive bacteria such as *Staphylococcus aureus* and *Streptococcus* spp. [28,29]. Due to the increasing interest in phytogenic alternatives to antibiotics in aquaculture, AND was selected for this study based on its dual potential as an antimicrobial [28] and immune modulator [30]. Moreover, its natural origin, low toxicity, and compatibility with nanocarrier systems make it a promising candidate for sustainable disease management in tilapia aquaculture [27]. Accordingly, advanced drug delivery systems are required to improve the bioavailability, stability, and therapeutic efficacy of AND in aquatic species.

Nanostructured lipid carriers (NLCs) have emerged as promising delivery systems for hydrophobic and unstable bioactives such as AND [31,32]. NLCs encapsulate bioactive compounds within a lipid matrix, protecting them from degradation, enabling controlled release, and enhancing absorption and therapeutic performance. Our previous research successfully applied NLCs for delivering 17α-methyltestosterone in red tilapia [33,34], clove oil-NLCs for whiteleg shrimp [35], and nano emulsified mangosteen peel extract in Nile tilapia [36]. However, no study has yet investigated the application of AND-loaded NLCs (AND-NLCs) in Nile tilapia. Therefore, this study aimed to formulate and characterize AND-NLCs, investigate their physicochemical properties, evaluate their in vitro antibacterial activity, and determine their in vivo effects on growth performance, feed utilization, hepatosomatic index, length–weight relationship, and disease resistance against *S. agalactiae* ENC06 in Nile tilapia. The decision to focus on *S. agalactiae* was based on its widespread prevalence and economic significance in Thai tilapia aquaculture, where serotype Ia strains—including ENC06—are commonly implicated in streptococcosis outbreaks. As such, it represents a high-priority pathogen for therapeutic intervention and vaccine development in this region.

## 2. Materials and Methods

### 2.1. Chemicals

Andrographolide was procured from Specialty Natural Product Co., Ltd. (Bangkok, Thailand). Sorbitan oleate, cetearyl alcohol, cocoglucoside, polyoxyethylene [20] sorbitan monolaurate, poloxamer, and glycerol were obtained from Croda (Thailand) Co., Ltd. (Bangkok, Thailand). Ethoxydiglycol was supplied by Myskin Recipe Co., Ltd. (Bangkok, Thailand). All other chemicals used were of analytical grade.

### 2.2. Preparation of Andrographolide-Loaded Nanostructured Lipid Carriers

AND was incorporated into NLC using the phase-inversion composition technique combined with high-energy homogenization [34]. Initially, the lipid phase had been prepared by dissolving 200 mg of AND in ethoxydiglycol (3.0 g), sorbitan oleate (3.0 g), and a mixture of cetearyl alcohol and cocoglucoside (1.0 g) under magnetic stirring at 60 °C. Simultaneously, the aqueous phase had been prepared by dissolving polyoxyethylene [20] sorbitan monolaurate (3.0 g), poloxamer 188 (2.0 g), and glycerol (2.5 g) in deionized water at 60 °C. A pre-emulsion was subsequently formed by combining the lipid and aqueous phases under continuous magnetic stirring at 500 rpm for 5 min at 60 °C. The final AND-NLC formulation was obtained using a probe sonicator set at 30% amplitude for 5 min (30-s pulses with 10-s intervals), followed by cooling to room temperature for subsequent analysis.

### 2.3. Particle Characterization

The hydrodynamic diameter, polydispersity index (PDI), and zeta potential of the synthesized nanoparticles were measured using dynamic light scattering (DLS) with a Zetasizer NanoZS (Malvern Instruments, Worcestershire, UK). Prior to measurement, the nanoparticle samples had been diluted 50-fold with deionized water and analyzed at 25 °C. All measurements were conducted in triplicate to ensure accuracy and reproducibility.

In addition, particle size and morphology were evaluated using a transmission electron microscope (TEM; HT7800, Hitachi, Tokyo, Japan) operated at 80 kV. Samples had been diluted 50-fold in deionized water (pH 7.0), placed onto carbon-coated grids, and dried using filter paper. The grids were stained with a uranyless staining solution for 2 min, rinsed with deionized water, and air-dried before imaging to assess morphological characteristics.

### 2.4. Encapsulation Efficiency and In Vitro Release of AND-NLC

The encapsulation efficiency (EE) of AND and AND-NLC was determined using a regenerated cellulose centrifugal filter device with a 30 kDa molecular weight cutoff (Amicon Ultra-15, Merck Millipore Ltd., Darmstadt, Germany) [34]. Approximately 1.5 mL of each sample was loaded into the filter unit and centrifuged according to the manufacturer’s instructions to separate unencapsulated AND. The filtrate containing the free drug was subjected to solvent extraction, following the method by Yostawonkul et al. [33]. The recovered AND was then passed through a 0.2 μm nylon syringe filter prior to analysis by high-performance liquid chromatography (HPLC) equipped with a photodiode array detector (HPLC-PAD 2998 separation unit and PDA, Water, Milford, MA, USA). Chromatographic separation was achieved using an Ascentis C18 reverse-phase column (5 μm, 25 cm × 4.6 mm), with a mobile phase of acetonitrile and methanol (60:40) at a flow rate of 1 mL/min. The injection volume was 20 μL, and the detection was carried out at a wavelength of 240 nm with a run time of 15 min. Encapsulation efficiency was calculated using the following formula:EE (%) = ((Ci − Cf)/Ci) × 100(1)
where *Ci* represents the initial concentration of AND used during formulation, and *Cf* denotes the amount of unencapsulated (free) AND present in the solution.

The in vitro release behavior of AND from the NLC formulation was evaluated in phosphate-buffered saline (PBS, pH 6.8) at 28 °C, based on modified protocols from previous studies [34]. A 1.5 mL sample of AND-NLC was placed in dialysis tubing (MWCO 3.5 kDa; Merck Millipore, Burlington, MA, USA) and immersed in 30 mL of receiving medium composed of PBS and ethanol (95:5 *v*/*v*). The dialysis system was maintained in a shaker incubator at 200 rpm (Vision Scientific Co., Yuseong-Gu, Daejeon-Si, Republic of Korea). At predetermined time intervals (1, 2, 3, 4, 5, 6, 7, 8, 10, 24, and 48 h), 1 mL of the external buffer was withdrawn and immediately replaced with fresh medium. The concentration of AND in the release medium was quantified using HPLC under the same analytical conditions as previously described. To evaluate the release mechanism, the data were fitted to the Avrami model (Equation (2)):*R* = 1 − exp (−(kt)^n^)(2)
where *R* is the cumulative release of AND at time *t*, *k* is the release rate constant, and *n* is the release mechanism parameter.

### 2.5. Fourier-Transform Infrared (FTIR) Spectroscopy Analysis

FTIR spectra of AND and AND-NLC were recorded using a Nicolet Summit Pro spectrometer (Thermo Scientific, Waltham, MA, USA) [37]. For sample preparation, AND or AND-NLC powder was mixed with potassium bromide (KBr) at a 1:100 weight ratio, finely ground, and compressed into translucent pellets. Spectral acquisition was conducted over the range of 500–4000 cm^−1^ with a resolution of 4 cm^−1^. A total of 64 scans were acquired and averaged for each sample. All measurements had been performed in triplicate to ensure reliability and reproducibility of the results.

### 2.6. Evaluation of Antimicrobial Activity

The antimicrobial properties of the AND or AND-NLC were evaluated using the agar well diffusion method. Given the focus of this study on targeted disease management, three *S. agalactiae* strains (FNA07, FPrA02, ENC06)—which had been previously isolated from diseased tilapia in Thailand—were selected as representative pathogenic strains for in vitro evaluation. These strains were cultured on blood agar plates (tryptic soy agar supplemented with 5% sheep blood; Bacto™, Waltham, MA, USA) and incubated at 28 °C for 24 h [9]. Bacterial suspensions were prepared in sterile 0.85% NaCl and adjusted to match a 0.5 McFarland standard. The suspensions were uniformly spread onto Mueller–Hinton agar (MHA) plates containing 5% sheep blood. Wells with a diameter of 6 mm were created using sterile pipette tips, and 50 µL of either AND or AND-NLC was added to each well. The plates were subsequently incubated at 28 °C for 24 h, after which the inhibition zone diameters (IZDs) were measured using a vernier caliper.

### 2.7. Evaluation of Minimum Inhibitory Concentration (MIC) and Minimum Bactericidal Concentration (MBC)

The MIC was determined using the broth microdilution technique [37]. AND or AND-NLC were serially diluted two-fold in Mueller–Hinton broth (MHB) across 96-well microplates to obtain ten concentration levels ranging from 2000 to 3.9 ppm, with a final volume of 100 µL per well. An equal volume (100 µL) of bacterial suspension, adjusted to 10^6^ CFU/mL, was added to each well. Plates were incubated at 28 °C for 24 h. Sterile MHB served as the negative control, while MHB with bacterial inoculum was used as the positive control. The MIC was recorded as the lowest concentration at which no visible bacterial growth was observed. For MBC determination, samples from each well were plated on Mueller–Hinton agar (MHA), and the absence of bacterial colonies after incubation was considered indicative of bactericidal activity.

### 2.8. Feed Supplementation

A concentration of 2 mg/mL of AND or AND-NLC was used to prepare the experimental diets using commercial tilapia feed (CP-7710; Charoen Pokphand Foods Public Company Limited, Samut Sakhon, Thailand). The feed preparation followed the method previously described by Kamble et al. [21], with minor modifications. Briefly, 1 g of commercial feed pellets was placed into a sterile 50 mL beaker. Then, 1 mL of either AND-NLC, free AND, or blank NLC was added, and the mixture was thoroughly stirred with a sterile stainless-steel spatula to ensure uniform coating. The control diet was prepared by mixing feed with distilled water. The treated feed was left to dry at room temperature overnight and was subsequently stored in clean plastic bags at 4 °C until further use.

### 2.9. Experimental Design

Nile tilapia fingerlings (5–7 g; total n = 600) were disinfected with 50 ppm formalin and were acclimated for 14 days in fiberglass tanks supplied with flow-through dechlorinated tap water and continuous aeration. The experiment was conducted using a completely randomized design with four treatment groups: control, AND, AND-NLC, and NLC. Each treatment included 50 fish per tank in triplicate. To maintain water quality, 30–50% of the water was exchanged daily. Throughout the acclimation and experimental periods, water temperature, dissolved oxygen, and pH were monitored daily and were maintained at 25–28 °C, 5.24–5.98 mg/L, and 7.48–8.16, respectively. Fish were fed commercial pellet feed twice daily at 3% of their body weight for a duration of two months. This design enabled direct comparison of AND and AND-NLC efficacy in improving fish performance and resistance against a relevant bacterial pathogen, without introducing interspecies or inter-pathogen variability.

### 2.10. Growth Performance and Feed Utilization

The growth performance and feed utilization efficiency of Nile tilapia were evaluated after 1 and 2 months of feeding [38]. At each time point, all fish from each treatment group were assessed by recording total weight and length. Growth parameters and feed utilization were calculated using the following formulas:Weight gain (WG, g/fish) = Final weight (FW, g) − Initial weight (IW, g)(3)Specific growth rate (SGR, %/day) = 100 × [(ln FW − ln IW)/Experimental days (T)](4)Mean daily intake (MDI, g/fish/day) = (Total feed intake (g)/T)/Number of fish per tank)(5)Feed conversion ratio (FCR) = Feed intake (g)/Wet weight gain (g)(6)Protein efficiency ratio (PER) = Wet weight gain (g)/Protein intake (g)(7)Hepatosomatic Index (HSI, %) = (Liver weight/Body weight) × 100(8)

### 2.11. Length–Weight Relationship, Growth Pattern, and Relative Condition Factor

The length–weight relationship (LWR) and relative condition factor (Kn) of Nile tilapia were evaluated after a 60-day feeding trial using experimental diets. The LWR, which describes the correlation between the fish’s total length and body weight, was calculated using the model proposed by Pauly [39]:*W* = aL^b^(9)

In this equation, W represents the body weight (g), L is the total length (cm), a denotes the intercept indicating the initial growth coefficient, and b represents the exponent that describes the growth pattern. To facilitate linear regression analysis, both sides of the equation were log-transformed:logW = loga + b·logL(10)

The exponent b characterizes the type of growth. A value of *b* = 3 indicates isometric growth, where the fish grows proportionally in length and weight. When *b* ≠ 3, growth is considered allometric—either positive (*b* > 3), meaning weight increases faster than length, or negative (*b* < 3), meaning length increases more than weight [40].

To assess the physiological well-being of the fish, the relative Kn was applied. This parameter provides insight into the health and robustness of individual fish relative to a standard weight–length relationship and was calculated using the formula from Le Cren [41]:Kn = Wo/Wc(11)

Here, Wo denotes the observed weight of the fish, and Wc is the expected weight calculated from the length–weight regression equation (*W = aL^b*). The regression was derived from all experimental fish at day 60 to represent the overall weight–length relationship. A Kn value greater than 1 indicates that the fish is in better-than-average condition, whereas a value less than 1 suggests suboptimal health or growth relative to a typical specimen of the same length.

### 2.12. Disease Resistance of Nile Tilapia Against S. agalactiace ENC06

ENC06 was selected for the challenge experiment based on its marginally higher zone of inhibition in the antibacterial assay and its documented association with severe streptococcosis outbreaks in Thai Nile tilapia aquaculture [16,17]. This strain had been widely recognized for its virulence and epidemiological relevance, making it a suitable model for evaluating disease resistance.

Following 60 days of feeding, Nile tilapia were challenged with *S. agalactiae* ENC06 by intraperitoneal (IP) injection of 100 µL bacterial suspension (1 × 10^6^ CFU/mL) into 30 fish per treatment group. Mortality was recorded over the subsequent 15 days. During the challenge period, water quality parameters were consistently maintained at 29.9 ± 0.05 °C for temperature, pH 7.37 ± 0.06, and dissolved oxygen at 4.97 ± 0.04 mg/L. To verify bacterial infection as the cause of mortality, tissues from dead or moribund fish—including kidney, spleen, and brain—were aseptically collected and cultured on tryptic soy agar (TSA) supplemented with 5% sheep blood. The plates were incubated at 28 °C for 24–48 h. Colonies were identified based on morphological features and β-hemolytic activity, confirming the presence of *S. agalactiae*. Cumulative mortality and relative percent survival (RPS) for the AND-NLC-supplemented groups were calculated using the following formulas [42]:Cumulative mortality (%) = (Number of mortalities/Total number of fish) × 100(12)RPS (%) = [1 − (Mortality (%) in treated group/Mortality (%) in control group)] × 100(13)

### 2.13. Statistical Analysis

Statistical analyses were performed using SPSS software (version 26; IBM Corp., Armonk, NY, USA). One-way analysis of variance (ANOVA), followed by Tukey’s Honest Significant Difference (HSD) test, was applied to determine significant differences among treatment groups for growth, feed utilization, and physiological parameters. Kaplan–Meier survival analysis was conducted to estimate cumulative survival, and log-rank (Mantel–Cox) tests were used to compare survival curves between groups. Additionally, the Cox proportional hazards model was employed to assess the relative risk of mortality (hazard ratios) among different dietary treatments. The Student’s *t*-test was used to evaluate differences between two time points (30 and 60 days). Pearson’s correlation coefficients (r) were calculated to examine relationships among nanoparticle characteristics, antibacterial activity, growth performance, and disease resistance. A *p*-value less than 0.05 was considered statistically significant.

## 3. Results

### 3.1. Physicochemical Properties

The AND-NLCs had exhibited an average particle size of 189.6 ± 3.2 nm, slightly larger than the blank NLCs (168.0 ± 2.2 nm), indicating successful incorporation of the drug (Table 1). The PDI had decreased from 0.262 ± 0.028 in NLCs to 0.159 ± 0.038 in AND-NLCs, reflecting enhanced particle uniformity. Zeta potential values had remained moderately negative (−20.0 to −21.33 mV), suggesting good colloidal stability. Notably, the AND-NLC formulation had demonstrated a high encapsulation efficiency of 90.58 ± 0.13%, confirming effective drug loading within the lipid matrix.

The size distribution by intensity of the AND-NLCs showed a peak around 100 nm (Figure 1A), indicating a well-defined particle size distribution suitable for drug delivery.

TEM images of AND-NLCs revealed spherical particles with an average size of approximately 200 nm (Figure 1B,C), consistent with the DLS data. Figure 1D shows the cumulative release profiles of AND and AND-NLCs, where AND-NLCs exhibited a controlled release pattern characterized by an initial burst followed by sustained release, indicating efficient drug encapsulation and controlled release behavior.

### 3.2. FTIR Analysis

FTIR spectra of AND and AND-NLC reveal distinct changes due to the encapsulation of andrographolide in nanostructured lipid carriers (Figure 2). The O-H stretching peak at 3330 cm^−1^ in AND had shifted slightly to 3340 cm^−1^ in AND-NLC, indicating minor interactions between AND and the lipid matrix. The C=O stretching peak at 1725 cm^−1^ for AND had shifted to 1741 cm^−1^ in AND-NLC, suggesting the formation of interactions between the drug and the lipid components. Additionally, the disappearance of the C=C stretching peak at 1674 cm^−1^ in AND-NLC further supported the encapsulation process, indicating modifications in the molecular environment.

The shifts in C-H bending and C-O stretching peaks from AND to AND-NLC (e.g., from 1314 cm^−1^ to 1343 cm^−1^) confirmed changes in the interaction between the drug and lipid carriers. These results indicated that the encapsulation of AND in NLCs had altered its molecular structure, suggesting improved stability and potential for enhanced bioavailability.

### 3.3. Antibacterial Activity

The antibacterial effects of AND and AND-NLC were evaluated against three *S. agalactiae* strains (FPrA02, FNA07, ENC06). The AND-NLC formulation had exhibited significantly larger inhibition zones (13–14 mm) compared to AND alone (6 mm for all strains), indicating enhanced antibacterial activity upon nanoencapsulation (Table 2).

However, MIC and MBC values for AND-NLC had remained >2000 ppm, suggesting that while AND-NLC had improved surface-level inhibition, it might require higher concentrations to inhibit or kill the bacteria in solution. In contrast, neither the control nor NLC had shown any detectable inhibition zones, MIC, or MBC values, confirming the absence of inherent antibacterial activity in the lipid carrier or the blank treatment.

### 3.4. Effect of Experimental Diets on Growth Performance and Feed Utilization of Nile Tilapia

After 30 days, no significant differences were observed in FW, WG, and SGR among the treatments, with all groups displaying comparable values (Figure 3A–C). However, by 60 days, fish fed with the AND-NLC diet had achieved the highest FW, WG, and SGR, significantly outperforming the control, AND, and NLC groups.

Feed intake did not differ significantly among the treatments at both 30 and 60 days (Figure 3D). Similarly, at 30 days, no significant differences were found in FCR and PER between the treatments (Figure 3E,F). By contrast, at 60 days, AND-NLC had exhibited the lowest FCR, indicating the most efficient feed conversion, followed by AND and NLC, with the control group exhibiting the highest FCR. Additionally, at 60 days, AND-NLC had shown the highest PER, significantly exceeding the control, NLC, and AND groups. The treatment and antibacterial activity (MIC and MBC) had strong positive effects on growth performance. Size and PDI also contributed positively, while the zeta potential appeared to have a detrimental effect when it was either too high or too low (Figure 4).

### 3.5. Effect of Time on Growth Performance and Feed Utilization of Experimental Diets

All treatment groups had shown significant increases in FW and WG from day 30 to day 60, indicating time-dependent improvement in growth. The SGR had increased significantly over time only in the NLC group, while it remained unchanged in the other groups. FI also had increased significantly in all groups over time, reflecting elevated metabolic demands associated with growth progression. Notably, FCR was significantly reduced in the AND-NLC group at 60 days, suggesting enhanced feed efficiency, whereas other groups showed no significant change. Similarly, PER had improved significantly across all treatments, with the most pronounced increase observed in the AND-NLC group. Detailed *t*-test results for each growth and feed utilization parameter are provided in Appendix A.

### 3.6. Effects of Experimental Diets on Hepatosomatic Index

At 30 days, no significant differences in HSI had been observed between treatments (Figure 5). However, at 60 days, significant differences emerged, with AND-NLC showing the highest HSI, significantly exceeding the values observed in the control, AND, and NLC groups, which did not differ significantly from each other. At 30 days, no significant differences had been detected among treatments. However, by day 60, the AND-NLC group exhibited significantly higher HSI values compared to all other treatments. Significant increases in HSI were also detected over time within the AND, AND-NLC, and NLC groups, suggesting enhanced liver mass potentially associated with improved nutrient metabolism or liver function. The detailed paired *t*-test results for HSI across time points are provided in Appendix A.

HSI is positively correlated with treatment and antibacterial activity (MIC and MBC), indicating improved liver health. In contrast, larger particle sizes and greater polydispersity (size, PDI) were negatively associated with HSI, while greater nanoparticle stability (zeta potential) was supportive of liver function (Figure 4).

### 3.7. Effects of Experimental Diets on Length–Weight Relationship and Growth Pattern

After 60 days of feeding, Nile tilapia fed diets supplemented with AND-NLC had exhibited the most favorable growth pattern among all treatments (Table 3). The regression coefficient (*b* = 3.350) in the AND-NLC group had indicated positive allometric growth, suggesting greater weight gain relative to length increase. This was significantly higher than the control (*b* = 2.697) and NLC (*b* = 2.619) groups, which had exhibited negative allometric growth, and also higher than the AND group (*b* = 3.036), which had shown moderate positive allometry. The correlation coefficient (r = 0.969) and coefficient of determination (*R*^2^ = 0.940) for the AND-NLC group had reflected a strong linear relationship between length and weight. All models were statistically significant (*p* < 0.001). The Kn was significantly higher in the AND group (1.229), indicating superior body condition, followed by the AND-NLC group (1.053). The NLC group had shown a significantly lower Kn value (0.971) compared to the other treatments. These results suggest that both AND and AND-NLC can improve fish condition, with AND-NLC demonstrating enhanced potential as a functional feed additive in tilapia aquaculture. The regression graphs illustrating the length–weight relationships for fish fed with control, AND, AND-NLC, and NLC diets are presented in Figure 6A–D.

Treatment and antibacterial activity (MIC and MBC) had improved LWR and Kn, promoting growth balance and fish health. In contrast, particles size and PDI had negatively impacted LWR and Kn, highlighting the need for particle size optimization. Zeta potential had supported both growth balance and fish health (Figure 4).

### 3.8. Effects of Experimental Diets on Survivorship of Nile Tilapia Following S. agalactiae Challenge (ENC06)

Kaplan–Meier survival curves demonstrated distinct differences in disease resistance among treatment groups following a 60-day dietary regimen and *S. agalactiae* challenge. A log-rank (Mantel–Cox) test comparing the cumulative survival percentages across the AND, AND-NLC, and NLC groups revealed significant differences (*X*^2^(4) = 78.973, *p* < 0.01) when compared to the control group (Figure 7A).

The infected control group had exhibited the highest mortality (77.8%) and lowest survival (22.2%), with an RPS of 0%, confirming high susceptibility to *S. agalactiae*. Among the supplemented groups, AND-NLC provided the greatest protective effect, with 26.7% mortality, 73.3% survival, and an RPS of 65.6% (Table 4).

The AND group also improved disease resistance compared to the control, with 35.6% mortality, 64.4% survival, and an RPS of 54.1%. In contrast, the NLC group offered minimal protection, with 75.5% mortality, 24.5% survival, and an RPS of only 2.5%, comparable to the infected control. These results indicated that AND-NLC supplementation substantially enhanced survival and disease resistance in Nile tilapia challenged with *S. agalactiae*, outperforming both free AND and the carrier alone. Survival and RPS are positively influenced by treatment and antibacterial activity (MIC and MBC), with zeta potential also contributing positively. Size and PDI were found to negatively affect survival, with larger particle sizes and broader particle size distributions being associated with lower survival rates (Figure 4).

### 3.9. Cox Regression Analysis of Treatment Effects on Survival Time

The Cox regression analysis revealed a significant overall effect of treatment on survival time (*X*^2^ = 69.427, df = 4, *p* < 0.001), confirming that survival outcomes varied among groups (Figure 7B). Compared to the reference group, the AND (Exp(B) = 0.190, *p* < 0.001), Control (Exp(B) = 0.256, *p* < 0.001), and NLC (Exp(B) = 0.102, *p* < 0.001) groups exhibited significantly higher hazard risks (i.e., lower survival probabilities). In contrast, the AND-NLC group (Exp(B) = 0.758, *p* = 0.253) did not show a statistically significant difference in hazard, suggesting a neutral effect on survival. These findings indicated that AND-NLC supplementation may have mitigated mortality risk more effectively than AND or NLC alone.

## 4. Discussion

The slight size increase from 168.0 to 189.6 nm confirmed successful drug loading into NLCs without compromising structural integrity. This was further supported by a reduced polydispersity index (from 0.262 to 0.159), indicating improved particle size uniformity [43]. Additionally, a moderately negative zeta potential (~−20 mV) suggested good colloidal stability, minimizing the risk of nanoparticle aggregation [44]. The high encapsulation efficiency (90.58%) demonstrated effective entrapment of the drug within the lipid matrix [45]. Consistent with these findings, the particle size distribution had peaked near 100 nm, and TEM images had revealed ~200 nm spherical particles, confirming uniform morphology. The drug release profile had shown an initial burst followed by sustained release [46,47]. Collectively, these physicochemical characteristics supported the functionality of lipid nanoparticles as efficient drug delivery systems [48].

Building on this, FTIR spectra revealed molecular-level changes upon encapsulating AND into NLCs [49]. A shift in the O-H stretching peak indicated interactions between AND and the lipid matrix, while a pronounced shift in the C=O peak and disappearance of the C=C peak in AND-NLC suggested stronger binding within the lipid environment. These alterations implied enhanced stability and bioavailability of AND [49]. By modifying its molecular structure, NLCs could protect AND from degradation and enhance its absorption, aligning with studies on nanoformulations that had improved poorly soluble drug bioavailability [49,50]. This is particularly important since AND has low oral bioavailability [51]. Thus, lipid-based systems offer a promising strategy to enhance phytoconstituent absorption [49].

Consistent with the structural and molecular findings, the antibacterial activity of AND-NLC was enhanced. While it showed larger inhibition zones, higher concentrations were required to achieve MIC and MBC in solution, a result commonly observed in nanoparticle-based drug delivery systems due to sustained release and delayed interactions with bacterial membranes [52]. AND, a labdane diterpenoid, exerts its antibacterial effect through multiple mechanisms. One of the primary modes of action involves disruption of bacterial membrane integrity, leading to leakage of intracellular contents and cell lysis [53]. AND has also been shown to inhibit the bacterial quorum sensing (QS) system, which plays a crucial role in virulence regulation, biofilm formation, and antibiotic resistance [54]. Furthermore, AND interferes with key bacterial metabolic pathways and inhibits DNA and protein synthesis [55]. These multimodal actions not only confer broad-spectrum antibacterial activity but also reduce the likelihood of resistance development. Fusidic acid-loaded lipid-core nanocapsules demonstrated similar efficacy against Gram-positive bacteria [56]. These results underscore the potential of NLCs in delivering antibacterial agents, particularly against resistant strains [57,58]. For example, cationic NLCs enhanced anti-MRSA activity with oxacillin [57], while virgin coconut oil NLCs exhibited potent effects against multidrug-resistant *S. aureus* [59].

These formulation advantages translated into biological outcomes. At 30 days, no significant differences in FW, WG, or SGR were observed among treatments. However, by 60 days, fish fed with AND-NLC exhibited significantly improved growth performance over the control, AND, and NLC groups. This delayed improvement reflected the sustained release and bioavailability of AND from NLCs [49,60], ensuring prolonged exposure and absorption [61]. Similar time-dependent responses were observed with curcumin nanoparticles in Nile tilapia (60 days) [62], chitosan nanoparticles (45 days) [63], and curcumin in Gilthead seabream (150 days) [64]. These findings support the hypothesis that extended exposure enhances nutrient absorption and growth [65]. Moreover, the correlation between particle size, PDI, and growth performance suggests that optimized nanoparticle characteristics improve nutrient uptake [66,67].

Beyond the sustained release, several complementary mechanisms contribute to the enhanced bioavailability and biological activity of AND-NLC in fish. Upon ingestion, NLCs are expected to release AND primarily in the intestinal tract, where the lipid matrix of the nanocarrier provides protection against acidic pH and enzymatic degradation in the stomach, ensuring a greater amount of intact compound reaches the absorption site [68]. The small particle size and low PDI enhance interaction with intestinal microvilli and promote uptake via transcellular and paracellular pathways [69]. Additionally, NLCs have been shown to improve intestinal permeability and epithelial absorption of lipophilic compounds, facilitating systemic circulation [70]. Some studies in fish suggest that lipid-based nanoparticles may also promote lymphatic transport, helping to bypass first-pass hepatic metabolism and increasing systemic bioavailability [71]. These mechanisms jointly support the delayed but significant improvements in growth and feed efficiency observed in the AND-NLC group after 60 days. Similar advantages have been reported with nano-curcumin [69], nano-omega-3 fatty acids [70], and *Laurus nobilis* essential oil nanoparticles [72] in Tilapia sp.

Feed intake did not differ significantly among treatments at 30 or 60 days. However, FI increased significantly over time, consistent with rising metabolic demands during fish growth [73]. The lack of treatment effects on FI suggests that AND-NLC improves feed utilization efficiency rather than stimulating appetite [74,75]. Similar findings were reported with nano-formulated omega-3 fatty acids [70] and herbal extract + nano quercetin [76], where FI remained unchanged while growth improved via enhanced nutrient absorption.

The feed conversion ratio followed a similar trend. No significant differences were observed at 30 days, but AND-NLC-fed fish had the lowest FCR at 60 days, indicating superior feed efficiency. This may result from improved gut health, nutrient absorption, and metabolic efficiency supported by sustained andrographolide release from NLCs [65,74,75]. Similar FCR improvements were reported using nanoemulsions from mangosteen peel [36], cinnamaldehyde [67], and *Lavandula officinalis* [66], and nano-curcumin [77] in Nile tilapia. Long-term feeding with nanocarriers allows bioactives to deliver sustained metabolic benefits. The FCR reduction in the AND-NLC group after 60 days confirms efficient feed utilization [78]. Correlations with antibacterial activity [79,80] and optimal particle size/PDI [63] further highlight the importance of controlled nanoparticle properties in promoting feed efficiency and disease resistance.

Protein efficiency ratio also improved by 60 days in the AND-NLC group. This was likely due to better protein utilization, supported by reduced catabolism and enhanced nitrogen retention via andrographolide’s anti-inflammatory and hepatoprotective effects [62,81,82,83]. Comparable PER improvements were observed with nano-omega-3 fatty acids [70], cinnamaldehyde nanoemulsion [67], and chitosan-vitamin C nanocomposite [82], attributed to antibacterial activity and nutrient absorption [70,74,84], confirming the role of formulation features in optimizing protein metabolism.

AND-NLC also significantly improved liver function, indicated by increased HSI over time. This enhancement is linked to the sustained release, bioavailability, and antibacterial properties of the formulation [83,85]. Herbal nanocarriers like those with curcumin [62] or andrographolide [86] have similarly outperformed free compounds in supporting liver health [87]. The correlation between HSI, NLC stability, and antibacterial activity emphasizes the relevance of zeta potential and size distribution in liver function outcomes [88,89].

By 60 days, fish in the AND-NLC group exhibited positive allometric growth, reflecting enhanced tissue deposition due to improved protein synthesis [90,91] and nutrient utilization [62,81]. Sustained release contributed to better absorption and feed efficiency [62,63,74]. Similar somatic growth patterns were reported in Nile tilapia fed guava and star gooseberry extract [22], silver barb and zebrafish fed moringa leaf meal [92], and plant protein-based diets [93]. Kn values above one in the AND and AND-NLC groups indicated optimal fish health, aligning with previous studies using guava and star gooseberry extract [22], sargassum meal [94], and varying maltose levels [95].

The AND-NLC supplementation significantly enhanced disease resistance in Nile tilapia challenged with *S. agalactiae* ENC06. This effect can be attributed to the sustained release, bioavailability, and antibacterial activity of andrographolide, with strong correlations to zeta potential and particle size. These traits prolong antibacterial effects and strengthen immune defense [96,97]. Similar findings have been reported with various herbal nanoformulations in fish [36,67,98,99]. Thus, optimizing carriers’ properties and bioactive selection is key to improving health outcomes in aquaculture systems.

In addition to the demonstrated growth and health benefits of AND-NLC supplementation, future studies should focus on elucidating the immunological pathways and molecular responses activated by this nanoformulated phytochemical. Transcriptomic or proteomic analyses could identify specific immune genes and signaling cascades involved in host defense. Furthermore, intestinal histology and gut microbiota profiling are recommended to better understand how AND-NLCs influence nutrient absorption and gut health. Long-term safety trials under commercial-scale conditions will also be critical to assess potential environmental impacts and cumulative toxicity. Finally, comparative studies with other nanoencapsulated phytocompounds may help identify synergistic combinations and refine dosing strategies for broader applications in aquaculture systems.

## 5. Conclusions

This study confirmed the potential of AND-NLCs to enhance drug delivery and support growth performance in Nile tilapia. The nanoformulation demonstrated favorable physicochemical properties, improved antibacterial activity, and greater bioavailability compared to free andrographolide. In vivo findings showed significant improvements in growth, feed conversion efficiency, protein utilization, liver condition, and survival following the *S. agalactiae* ENC06 challenge. These results support the application of AND-NLCs as a promising functional feed additive in aquaculture. Future research should investigate the immunological mechanisms of action, assess gut health and microbial dynamics, and evaluate long-term safety and scalability under real-world farming conditions.

## Figures and Tables

**Figure 1 animals-15-02117-f001:**
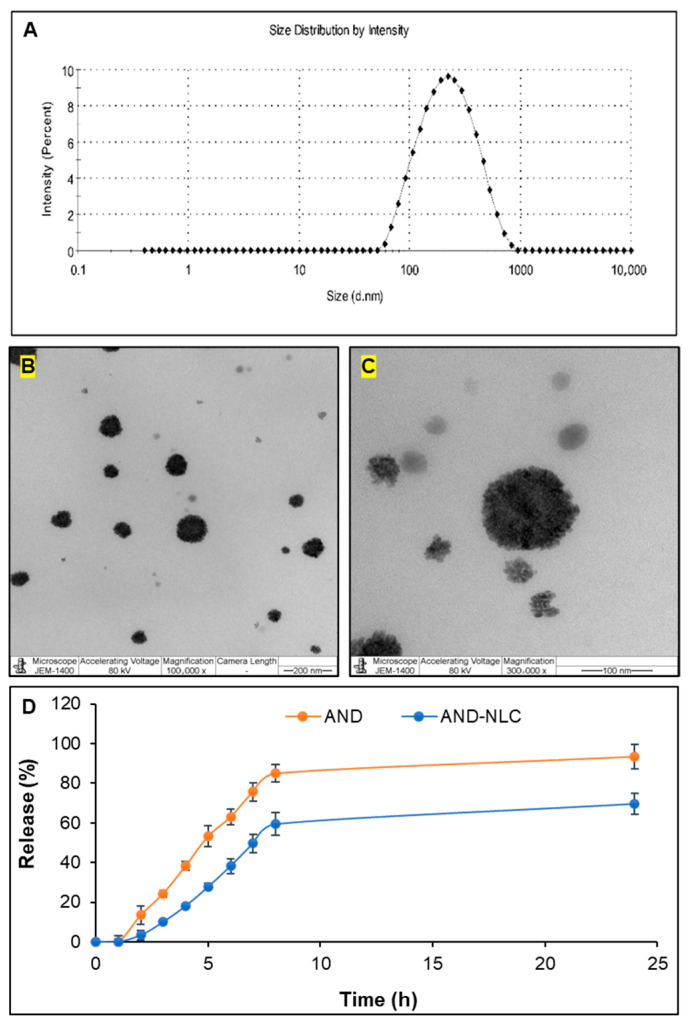
Characterization of AND-NLC. (**A**) Size distribution by intensity measured using dynamic light scattering (DLS). (**B**,**C**) Morphology of AND-NLC observed by transmission electron microscopy (TEM) at 200 nm and 100 nm scale bars, respectively. (**D**) In vitro release profile of andrographolide from NLCs over time.

**Figure 2 animals-15-02117-f002:**
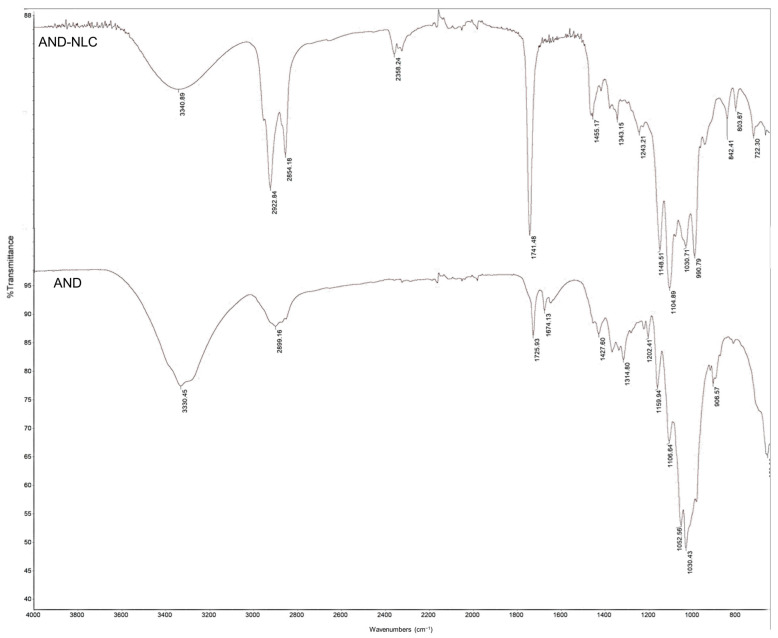
FTIR spectra of andrographolide (AND, bottom spectrum) and andrographolide loaded in nanostructured lipid carriers (AND-NLC, top spectrum). Characteristic shifts in functional group peaks confirm the interaction of AND with the lipid matrix, indicating successful encapsulation and potential enhancement in stability and bioavailability.

**Figure 3 animals-15-02117-f003:**
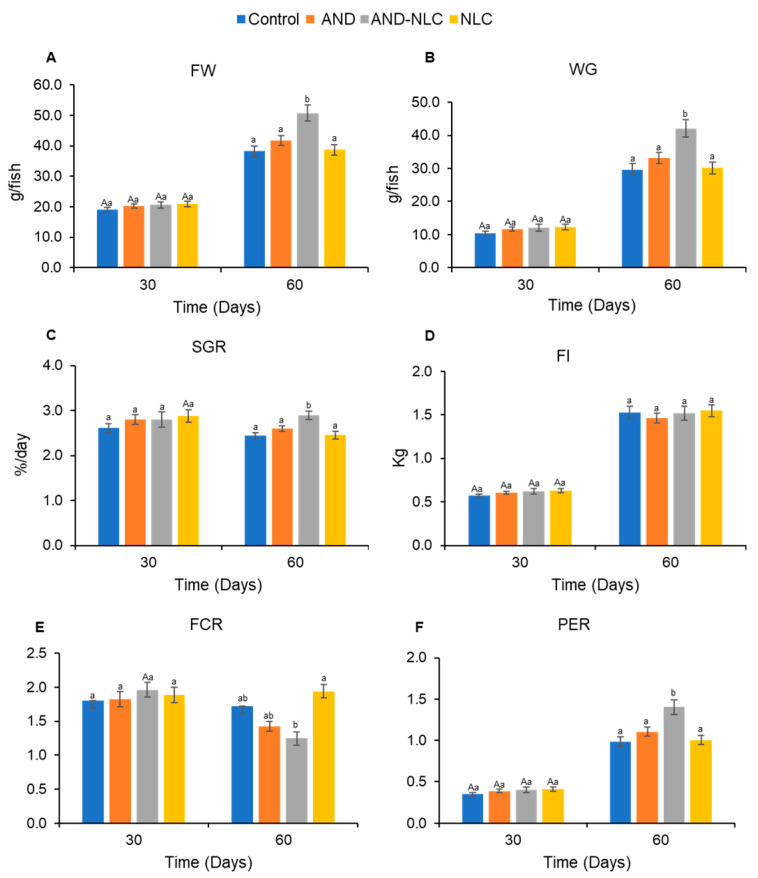
Growth performance and feed utilization parameters of Nile tilapia fed diets supplemented with control, andrographolide (AND), andrographolide-loaded nanostructured lipid carriers (AND-NLC), and nanostructured lipid carriers (NLC) for 30 and 60 days. (**A**–**C**) Final weight (FW), weight gain (WG), and specific growth rate (SGR), respectively. (**D**–**F**) Feed intake (FI), feed conversion ratio (FCR), and protein efficiency ratio (PER), respectively. Different lowercase superscripts indicate significant differences between treatments at each time point. Uppercase superscripts indicate significant differences over time between treatments.

**Figure 4 animals-15-02117-f004:**
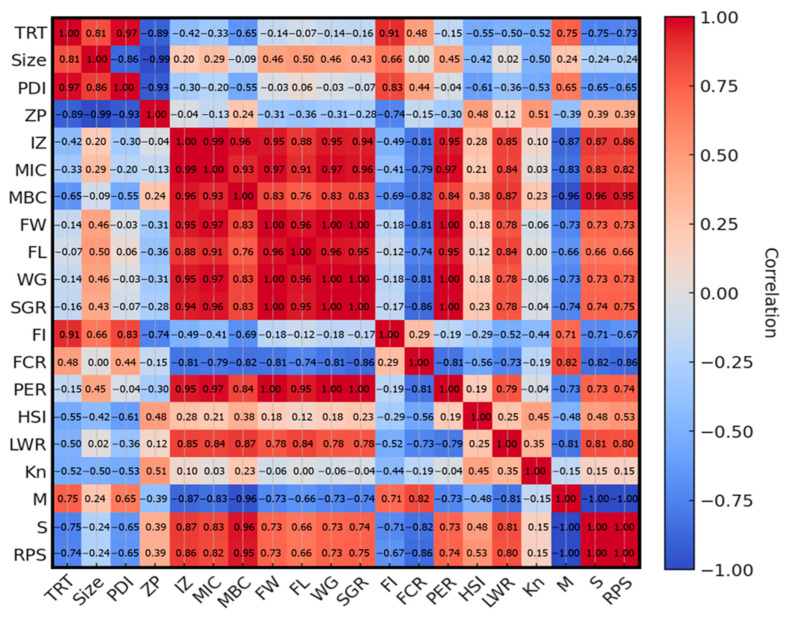
Pearson correlation heatmap showing relationships among nanoparticle characteristics, antibacterial activity, growth, feed utilization, physiological indices, and disease resistance parameters in Nile tilapia. Strong positive and negative correlations are indicated by color intensity and direction. Treatment (TRT); nanoparticle characteristics: particle size (size), polydispersity index (PDI), and zeta potential (ZP); antibacterial activity: inhibition zone (IZ), minimum inhibitory concentration (MIC), and minimum bactericidal concentration (MBC); growth performance: final weight (FW), final length (FL), weight gain (WG), and specific growth rate (SGR); feed utilization: feed intake (FI), feed consumption ratio (FCR), protein efficiency ratio (PER); physiological indices: hepatosomatic index (HSI), length–weight relationship (LWR), and relative condition factor (Kn); and disease resistance: mortality (M), survival (S), and relative percent survival (RPS) in Nile tilapia.

**Figure 5 animals-15-02117-f005:**
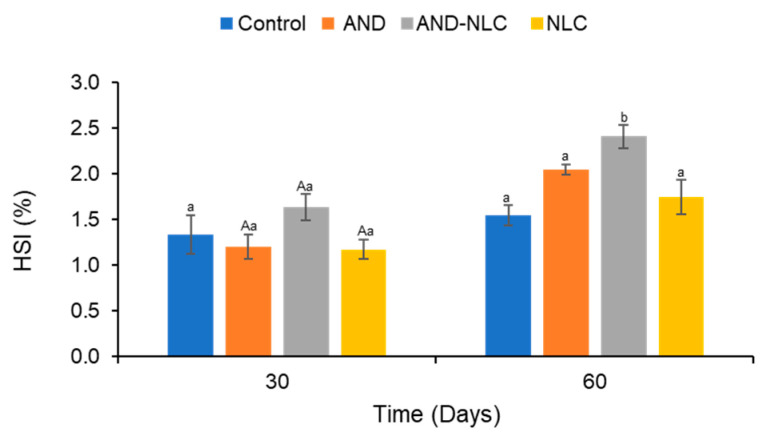
Hepatosomatic index (HSI) of Nile tilapia fed diets supplemented with control, andrographolide (AND), andrographolide-loaded nanostructured lipid carriers (AND-NLC) and nanostructured lipid carriers (NLC) for 30 and 60 days. Different lowercase superscripts indicate significant differences between treatments at each time point. Uppercase superscripts indicate significant differences over time between treatments.

**Figure 6 animals-15-02117-f006:**
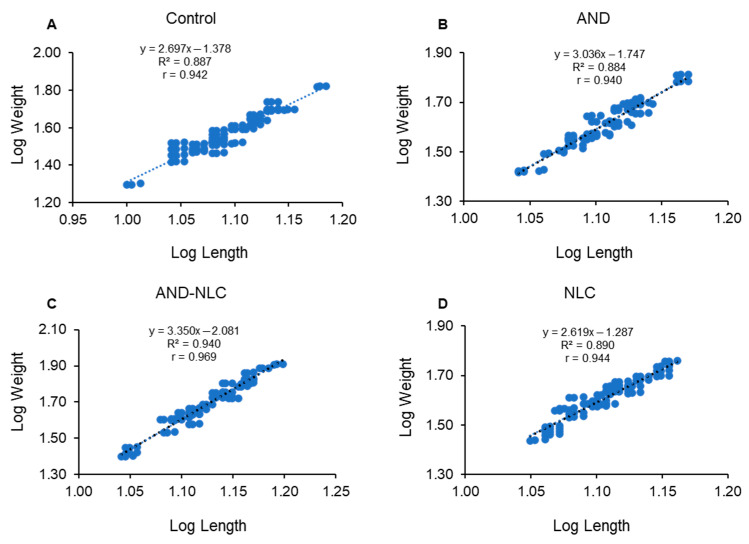
Log-transformed length–weight relationship and regression equations of Nile tilapia fed different diets for 60 days. (**A**–**D**) control, andrographolide (AND), andrographolide-loaded nanostructured lipid carriers (AND-NLC) and nanostructured lipid carriers (NLC).

**Figure 7 animals-15-02117-f007:**
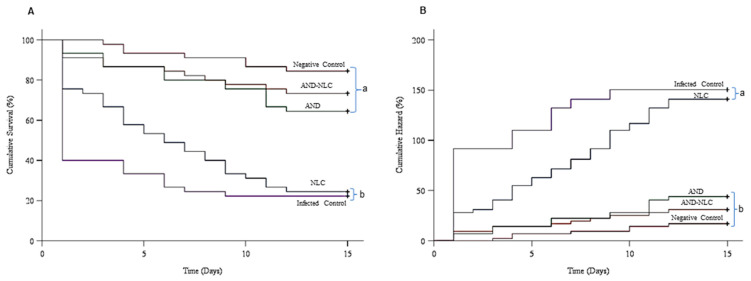
Disease resistance outcomes following *S. agalactiae* ENC06 challenge in Nile tilapia. (**A**) Kaplan–Meier survival curves showing cumulative survival across diet groups. (**B**) Cox proportional hazards model estimating relative mortality risk across treatments. Different lowercase superscripts indicate significant differences between treatments.

**Table 1 animals-15-02117-t001:** Physicochemical properties of andrographolide (AND) loaded in nanostructured lipid carriers (NLC).

Formulation	Size (nm)	Polydispersity Index	Zeta Potential (mV)	Encapsulation Efficiency (%)
NLC	168.0 ± 2.2	0.262 ± 0.028	−21.33 ± 1.0	ND
AND-NLC	189.6 ± 3.2	0.159 ± 0.038	−20.0 ± 0.5	90.58 ± 0.13

Values were the mean ± SD of triplicate samples. ND: not detected.

**Table 2 animals-15-02117-t002:** Inhibition zone diameter (IZD), minimum inhibitory concentration (MIC), and minimum bactericidal concentration (MBC) of andrographolide (AND) and andrographolide-loaded nanostructured lipid carriers (AND-NLC) against *S. agalactiae* strains (FPrA02, FNA07, ENC06).

Formulation	IZD (mm)			MIC (ppm)			MBC (ppm)		
	FPrA02	FNA07	ENC06	FPrA02	FNA07	ENC06	FPrA02	FNA07	ENC06
AND	6 ± 0 ^a^	6 ± 0 ^a^	6 ± 0 ^a^	1000	1000	1000	2000	2000	2000
AND-NLC	13 ± 2 ^b^	13 ± 2 ^b^	14 ± 1 ^b^	>2000	>2000	>2000	>2000	>2000	>2000

Values were the mean ± SD of triplicate samples. Different lowercase superscripts in each column indicate significant differences between treatments.

**Table 3 animals-15-02117-t003:** Length–weight relationship, growth pattern, and relative condition factor of Nile tilapia fed diets supplemented with control, andrographolide (AND), andrographolide-loaded nanostructured lipid carriers (AND-NLC), and nanostructured lipid carriers (NLC) for 60 days.

Parameters	Control	AND	AND-NLC	NLC
N	90	90	90	90
L_Min-Max_ (cm)	10.0–15	11.0–14.8	11.0–15.8	11.2–14.5
W_min-max_ (g)	19.8–66.4	26.1–65.0	25.0–81.2	27.3–57.4
a	−1.378	−1.747	−2.081	−1.287
*b*	2.697	3.036	3.350	2.619
SE (b)	0.103	0.117	0.090	0.095
CI (b)	2.493–2.901	2.803–3.269	3.170–3.530	2.424–2.813
r	0.942	0.940	0.969	0.944
R^2^	0.887	0.884	0.940	0.890
*p*	0.001	0.001	0.001	0.001
*t*-test sig	0.001	0.001	0.001	0.001
Growth behavior	Negative allometry	Positive allometry	Positive allometry	Negative allometry
K_n_	1.195 ^a^	1.229 ^a^	1.052 ^b^	0.971 ^c^
Min-Max	0.962–1.452	1.016–1.463	0.918–1.236	0.876–1.188
SE	0.010	0.011	0.008	0.013

N: sample size; L: length in centimeters; W: weight in grams; Min/Max: minimum and maximum values; a: intercept of the regression equation; b: slope; SE: standard error; CI (b): confidence interval for the slope; r: correlation coefficient; R^2^: coefficient of determination; *p*: statistical significance (considered significant at *p* < 0.05); Kn: relative condition factor. A *t*-test was used to assess whether the slope (b) significantly deviated from the expected value of 3. The growth pattern was interpreted based on the value of b. Different lowercase superscripts in the Kn row indicate significant differences between treatments.

**Table 4 animals-15-02117-t004:** The cumulative mortality and relative percent survival (RPS) of Nile tilapia fed diets supplemented with control, andrographolide (AND), andrographolide-loaded nanostructured lipid carriers (AND-NLC), and nanostructured lipid carriers (NLC) for 60 days after being challenged with *S. agalactiae* ENC06.

Treatments	Mortality (%)	Survival (%)	RPS (%)
Infected control	77.8 ± 2.2 ^a^	22.2 ± 2.2 ^a^	0.0
Negative control	15.5 ± 2.2 ^c^	84.5 ± 2.2 ^c^	80.0 ± 2.6 ^c^
AND	35.6 ± 4.4 ^b^	64.4 ± 4.4 ^b^	54.1 ± 6.4 ^b^
AND-NLC	26.7 ± 3.8 ^bc^	73.3 ± 3.8 ^bc^	65.6 ± 4.9 ^bc^
NLC	75.53 ± 2.2 ^a^	24.5 ± 2.2 ^a^	2.5 ± 0.8 ^a^

Values were the mean ± SD of triplicate samples. Different lowercase superscripts in each column indicate significant differences between treatments.

## Data Availability

The data presented in this study are available on request from the corresponding author.

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
