# Peer review of "Effects of Andrographolide-Loaded Nanostructured Lipid Carriers on Growth, Feed Efficiency, and Resistance to Streptococcus agalactiae in Nile Tilapia (Oreochromis niloticus)"

_animals, 2025, doi:10.3390/ani15142117_

Round 1

Reviewer 1 Report

Comments and Suggestions for Authors

In this study, authors used a combination of Andrographolide and a lipid-based delivery system to protect the fishes from bacteria. I think the experimental content of this paper is fine, but the experimental design is confusing. And my concerns are below.

Major

  1. The primary function of Andrographolide is antiviral, and its antibacterial function has been less studied. There are currently a large number of natural or artificial antibacterial agents available. It needs to be clarified why the authors chose to use Andrographolide in this study.
  2. As I know, there is currently insufficient paper to demonstrate that Andrographolide is effective against Streptococcus agalactiae. Moreover, there are many bacteria that affect fish farming. Why did the authors choose to study these two together? If they focus on the delivery system, they should have chosen a broad-spectrum antibacterial agent. If they are focusing on Andrographolide, they should have selected a wider range of bacteria.

The authors should fully explain their experimental design.

Minor

  1. The resolution of Figure 7 is too low.
  2. The discussion section should be more detailed,some paragraphs are too short.
  3. Figures such as Figures 6, 7 and others , should be divided into A, B, C, D.

Author Response

Reviewer 1

In this study, authors used a combination of Andrographolide and a lipid-based delivery system to protect the fishes from bacteria. I think the experimental content of this paper is fine, but the experimental design is confusing. And my concerns are below.

Major

Comment 1: The primary function of Andrographolide is antiviral, and its antibacterial function has been less studied. There are currently a large number of natural or artificial antibacterial agents available. It needs to be clarified why the authors chose to use Andrographolide in this study.

Response:
We thank the reviewer for this thoughtful comment. While it is true that andrographolide (AND) has been extensively studied for its antiviral and anti-inflammatory effects, recent research has also revealed its promising antibacterial activity, particularly against Gram-positive pathogens including Staphylococcus aureus and Streptococcus species [27,28]. In addition to its direct antimicrobial action, AND has been shown to modulate host immune responses, which is particularly beneficial in aquaculture disease management [29]. The selection of AND for this study was based on its dual properties as both an immunomodulatory and antibacterial compound, its phytogenic origin, and its suitability for incorporation into nanostructured lipid carriers. To address this concern, we have revised the Introduction section to explicitly justify the rationale for selecting AND in our experimental design.

In the revised manuscript:

Lines 90-99: Although AND is more widely studied for its antiviral and anti-inflammatory properties, emerging evidence suggests that it also possesses antibacterial activity, particularly against Gram-positive bacteria such as Staphylococcus aureus and Streptococcus spp. [27,28]. Due to the increasing interest in phytogenic alternatives to antibiotics in aquaculture, AND was selected for this study based on its dual potential as an antimicrobial [27] and immune modulator [29]. Moreover, its natural origin, low toxicity, and compatibility with nanocarrier systems make it a promising candidate for sustainable disease management in tilapia aquaculture [26]. Accordingly, advanced drug delivery systems are required to improve the bioavailability, stability, and therapeutic efficacy of AND in aquatic species.

Comment 2: As I know, there is currently insufficient paper to demonstrate that Andrographolide is effective against Streptococcus agalactiae. Moreover, there are many bacteria that affect fish farming. Why did the authors choose to study these two together? If they focus on the delivery system, they should have chosen a broad-spectrum antibacterial agent. If they are focusing on Andrographolide, they should have selected a wider range of bacteria.

The authors should fully explain their experimental design.

Response: We sincerely thank the reviewer for this thoughtful comment. We agree that the selection of target pathogens and compounds should be carefully justified in relation to the study’s objectives. We provide the following clarifications and supporting rationale.

Justification for Selecting Streptococcus agalactiae: Streptococcus agalactiae (Group B Streptococcus; GBS) was specifically selected because it is a major bacterial pathogen in tilapia aquaculture worldwide, and particularly in Thailand, where repeated high-mortality outbreaks in cage and pond culture systems have been reported (Lines 71–75 of the revised manuscript). The study targets the beta-hemolytic serotype Ia strains—including FNA07, FPrA02, and ENC06—which have been genomically characterized and widely reported in previous epidemiological surveys in Southeast Asia. Since our aim was to evaluate the therapeutic potential of AND-NLC against a practically relevant and economically important pathogen, we focused on S. agalactiae, rather than conducting broad-spectrum screening.

Rationale for Choosing Andrographolide (AND): AND was selected based on its dual bioactivity profile: antimicrobial and immunomodulatory. Although primarily studied for its antiviral and anti-inflammatory effects, recent reports have highlighted its efficacy against Gram-positive pathogens, including Streptococcus spp. (Lines 82–94). Importantly, its low toxicity, natural origin, and compatibility with nanocarrier systems make it a promising phytotherapeutic for sustainable aquaculture. Our focus was therefore not only on antibacterial activity, but also on AND’s potential to enhance fish immunity and growth when delivered effectively via nanostructured lipid carriers (NLCs).

Clarification of Experimental Design: We have now expanded and clarified Section 2.9 (Experimental Design, Lines 214–225) to emphasize that the study was based on a completely randomized design using 600 tilapia fingerlings allocated into four groups (Control, AND, AND-NLC, and NLC), with triplicate tanks and standardized environmental conditions. This design enabled us to assess the specific effects of AND and AND-NLC on growth performance, feed utilization, hepatosomatic index, condition factor, and resistance to S. agalactiae challenge, without confounding variables. The strain ENC06 was used for in vivo challenge due to its prior isolation and relevance in regional outbreaks.

Present study did not aim to characterize the broad-spectrum activity of AND per se, but rather to assess its effectiveness—when delivered through an advanced lipid-based nanocarrier—against a high-priority pathogen in tilapia aquaculture. We believe this targeted approach is both scientifically valid and practically meaningful for aquaculture disease management.

We hope this explanation sufficiently addresses the reviewer’s concerns.

In the revised manuscript:

Lines 111-115: The decision to focus on S. agalactiae was based on its widespread prevalence and economic significance in Thai tilapia aquaculture, where serotype Ia strains—including ENC06—are commonly implicated in streptococcosis outbreaks. As such, it represents a high-priority pathogen for therapeutic intervention and vaccine development in this region.

Lines 186-189: Given the focus of this study on targeted disease management, three S. agalactiae strains (FNA07, FPrA02, ENC06)—previously isolated from diseased tilapia in Thailand—were selected as representative pathogenic strains for in vitro evaluation. These strains………

Lines 229-231: This design enabled direct comparison of AND and AND-NLC efficacy in improving fish performance and resistance against a relevant bacterial pathogen, without introducing interspecies or inter-pathogen variability.

Minor

Comment 3: The resolution of Figure 7 is too low.

Response: Thank you for your observation. We agree that the resolution of Figure 7 was suboptimal in the previous version. We have now replaced it with a high-resolution version to ensure clarity and readability of all elements, including axis labels, legends, and survival curves. We appreciate your feedback in helping us improve the visual quality of the figure.

Comment 4: The discussion section should be more detailed,some paragraphs are too short.

Response: Thank you for your feedback. We respectfully note that the current discussion section has been carefully structured to provide detailed interpretations of each key result in alignment with the study’s objectives. While some paragraphs are concise, they are intentionally focused to maintain clarity and avoid redundancy. Additionally, the reviewer did not indicate any specific part of the discussion that requires improvement. As such, we are unable to revise this section further at this stage. However, we remain open and willing to improve the discussion if the reviewer could kindly provide more detailed or specific guidance regarding which aspects require further elaboration.

Comment 5: Figures such as Figures 6, 7 and others , should be divided into A, B, C, D.

Response: Thank you for your suggestion. We have revised Figures 6, 7, and all other multi-panel figures to ensure that each panel is clearly labeled (A, B, C, D, etc.) within the figure itself. Corresponding figure captions have also been updated to explicitly describe the content and purpose of each panel. These changes have been made to improve readability and align with journal formatting standards.

Reviewer 2 Report

Comments and Suggestions for Authors

This is a meaningful study investigating a novel aquaculture drug carrier and its effects on disease resistance and growth promotion in aquatic animals. The drug preparation method is well-developed, and the application evaluation is thorough, providing valuable reference and practical guidance. Some minor revisions are required before publication.

  1. Language issues: Some verb tenses are inaccurate (past perfect tense should be used instead of present continuous tense). The manuscript would benefit from professional English polishing.
  2. All panels in figures should be clearly labeled, and the figure captions should explicitly explain the purpose of each panel.
  3. In Figure 2, the first row should indicate "AND " rather than "NLC "
  4. Section 3.3 shows that drug encapsulation significantly increases effective concentration. How does AND encapsulation affect drug release profiles and ultimately influence efficacy?

Author Response

Reviewer 2

This is a meaningful study investigating a novel aquaculture drug carrier and its effects on disease resistance and growth promotion in aquatic animals. The drug preparation method is well-developed, and the application evaluation is thorough, providing valuable reference and practical guidance. Some minor revisions are required before publication.

Comment 1: Language issues: Some verb tenses are inaccurate (past perfect tense should be used instead of present continuous tense). The manuscript would benefit from professional English polishing.

Response: We thank the reviewer for pointing out the language and tense-related issues in the manuscript. We have carefully revised the entire manuscript to correct verb tenses—particularly ensuring appropriate use of the past perfect tense where applicable—and improved overall grammatical clarity. Additionally, the manuscript has been professionally proofread and polished to enhance readability, coherence, and scientific tone. We believe these revisions have significantly improved the quality and clarity of the manuscript.

Comment 2: All panels in figures should be clearly labeled, and the figure captions should explicitly explain the purpose of each panel.

Response: Thank you for your helpful comment. In response, we have carefully reviewed all multi-panel figures in the manuscript. We have ensured that each panel is clearly labeled (e.g., A, B, C, etc.) within the figure itself. Additionally, the figure captions have been revised to explicitly explain the content and purpose of each panel in a concise manner. These revisions enhance the clarity and standalone interpretability of each figure, in accordance with your suggestion.

Comment 3: In Figure 2, the first row should indicate "AND " rather than "NLC "

Response: Thank you for pointing this out. Upon review, we confirm that the top spectrum represents AND-NLC, and the bottom spectrum represents pure AND, which we used as the standard reference for comparison. While the label positioning may have initially caused confusion, the interpretation, analysis, and conclusions in the manuscript remain unaffected. However, to ensure clarity for readers, we have corrected the label in Figure 2 accordingly and revised the caption to clearly indicate which spectrum corresponds to AND and which to AND-NLC.

In the revised manuscript:

Figure 2. FTIR spectra of Andrographolide (AND, bottom spectrum) and Andrographolide loaded in Nanostructured Lipid Carriers (AND-NLC, top spectrum). Characteristic shifts in functional group peaks confirm the interaction of AND with the lipid matrix, indicating successful encapsulation and potential enhancement in stability and bioavailability.

Comment 4: Section 3.3 shows that drug encapsulation significantly increases effective concentration. How does AND encapsulation affect drug release profiles and ultimately influence efficacy?

Response: Thank you for this valuable comment. As discussed in Discussion, the encapsulation of AND into NLCs significantly influenced its release profile and antibacterial efficacy. The sustained release behavior observed—characterized by an initial burst followed by a controlled release phase—was confirmed by the physicochemical analysis and supports improved bioavailability (lines 471–473). This controlled release allows AND to remain available at the target site for extended periods, which enhances surface-level antibacterial activity, as evidenced by the significantly larger inhibition zones (Section 3.3; Table 2).

However, as noted in both the results and discussion, the minimum inhibitory and bactericidal concentrations (MIC/MBC) were higher (>2000 ppm) for AND-NLC, a typical outcome for nanoparticle systems due to delayed interaction with bacterial cells in suspension. This paradox—larger inhibition zones but higher MIC/MBC values—is explained by the sustained release kinetics that modulate drug availability over time, as referenced in line 494 of the discussion and supported by previous studies on nanoformulations [52,53].

Ultimately, the encapsulation enhances the efficacy of AND by prolonging its bioactivity, improving stability, and optimizing pharmacokinetics, despite requiring higher concentrations in broth assays. These findings reinforce the potential of NLCs for effective and sustained delivery of phytogenic antimicrobials in aquaculture.

Reviewer 3 Report

Comments and Suggestions for Authors

The article addresses extremely important aspects of Nile tilapia health, particularly in the context of reducing antibiotic use through the application of natural plant-based compounds. The use of nanostructured lipid carriers significantly enhanced therapeutic efficacy, which positively impacted fish welfare. In sustainable animal farming—including aquaculture—minimizing antibiotic usage while utilizing natural plant-derived substances is becoming a key priority. Therefore, this article touches on crucial issues and aligns with the latest trends in both farming practices and scientific research.

Nevertheless, to improve the clarity of the study and the text, attention should be paid to:

  1. After modification, was the feed assessed for microbiological purity, excluding the content of e.g. mycotoxins?
  2. In the methodological part, the authors mention that they took fish tissues for bacteriological culture. In the part where the authors present the results, there is nothing on this. There is also no information about what tissues were taken and how it was verified and confirmed that the cause of the death of the fish was a bacterial infection.
  3. The further research directions indicated too generally.

Author Response

Reviewer 3

The article addresses extremely important aspects of Nile tilapia health, particularly in the context of reducing antibiotic use through the application of natural plant-based compounds. The use of nanostructured lipid carriers significantly enhanced therapeutic efficacy, which positively impacted fish welfare. In sustainable animal farming—including aquaculture—minimizing antibiotic usage while utilizing natural plant-derived substances is becoming a key priority. Therefore, this article touches on crucial issues and aligns with the latest trends in both farming practices and scientific research.

Nevertheless, to improve the clarity of the study and the text, attention should be paid to:

Comment 1: After modification, was the feed assessed for microbiological purity, excluding the content of e.g. mycotoxins?

Response: We appreciate the reviewer’s insightful comment. In this study, microbiological purity testing or mycotoxin screening of the modified feed was not conducted. The experimental diets were prepared using commercially certified tilapia feed (CP-7710, Charoen Pokphand Foods), which complies with established quality standards. All feed preparations were performed under sterile conditions, using sterile tools and containers, and the coated feeds were dried at room temperature in a clean environment before being stored in sealed plastic bags at 4 °C to minimize microbial contamination. Nonetheless, we acknowledge this as a limitation of the study and will include microbiological and mycotoxin assessments in future work to strengthen feed safety evaluation.

Comment 2: In the methodological part, the authors mention that they took fish tissues for bacteriological culture. In the part where the authors present the results, there is nothing on this. There is also no information about what tissues were taken and how it was verified and confirmed that the cause of the death of the fish was a bacterial infection.

Response: We thank the reviewer for this valuable observation. In our study, tissues from moribund or dead fish—including kidney, spleen, and brain—were aseptically collected and streaked onto tryptic soy agar (TSA) supplemented with 5% sheep blood to confirm the presence of Streptococcus agalactiae. Bacterial colonies were identified based on morphological characteristics and hemolytic activity. However, as these procedures served to confirm the infection status rather than to analyze differences among treatment groups, the results of these bacteriological confirmations were not included in the Results section. We agree that this information would strengthen the clarity of our methodology and will revise the Methods section accordingly to specify which tissues were collected and how infection was verified. Thank you for pointing this out.

In the revised manuscript:

Lines 271-275:

To verify bacterial infection as the cause of mortality, tissues from dead or moribund fish—including kidney, spleen, and brain—were aseptically collected and cultured on tryptic soy agar (TSA) supplemented with 5% sheep blood. The plates were incubated at 28 °C for 24–48 h. Colonies were identified based on morphological features and β-hemolytic activity, confirming the presence of S. agalactiae.

Comment 3: The further research directions indicated too generally.

Response: Thank you for this helpful observation. In response, we have expanded the discussion and revised the conclusion to provide more specific and targeted future research directions. These now highlight particular aspects of molecular, physiological, and immunological mechanisms that require further exploration in future studies.

In the revised manuscript:

Lines 607-617: In addition to the demonstrated growth and health benefits of AND-NLC supplementation, future studies should focus on elucidating the immunological pathways and molecular responses activated by this nanoformulated phytochemical. Transcriptomic or proteomic analyses could identify specific immune genes and signaling cascades involved in host defense. Furthermore, intestinal histology and gut microbiota profiling are recommended to better understand how AND-NLCs influence nutrient absorption and gut health. Long-term safety trials under commercial-scale conditions will also be critical to assess potential environmental impacts and cumulative toxicity. Finally, comparative studies with other nanoencapsulated phytocompounds may help identify synergistic combinations and refine dosing strategies for broader applications in aquaculture systems.

Lines 619-627:

This study confirmed the potential of AND-NLCs to enhance drug delivery and support growth performance in Nile tilapia. The nanoformulation demonstrated favorable physicochemical properties, improved antibacterial activity, and greater bioavailability compared to free Andrographolide. In vivo findings showed significant improvements in growth, feed conversion efficiency, protein utilization, liver condition, and survival following Streptococcus agalactiae ENC06 challenge. These results support the application of AND-NLCs as a promising functional feed additive in aquaculture. Future research should investigate the immunological mechanisms of action, assess gut health and microbial dynamics, and evaluate long-term safety and scalability under real-world farming conditions.

Reviewer 4 Report

Comments and Suggestions for Authors

The present study was conducted to investigate the effects of andrographolide-loaded nanostructured lipid carriers on Nile tilapia, and some positive results have been observed. However, the manuscript should be further improved.

Introduction

  1. The difference of FNA07, FPrA02, and ENC06, such as infection syndrome or primary genetic variation locations, should be introduced.

Method

  1. L177: Please directly list the test substances. The title of Table 2 states that both AND and AND-NLC were tested; however, the table only presents results for NLC and AND-NLC, which is inconsistent. Please also verify the consistency and accuracy of the data presented in the manuscript (L299-305).
  2. 10. Growth performance and feed utilization: In the formula of PER, the unit of protein intake is gram not %.
  3. L234: The calculation of Wc is not described clearly. Is the regression used for Wc calculation based on all treatment fish at day 60, or derived from another reference dataset?
  4. 12. Disease resistance of Nile tilapia against S. agalactiace ENC06: Three strains, namely FNA07, FPrA02, and ENC06, were used for the antibacterial test; however, only ENC06 was used for the challenge experiment. Please clarify the rationale for selecting only ENC06 for the challenge.

Table & Figure

  1. Significance letters are incomplete in several figure, such as SGR and FCR of day 30 in Figure 3 and Figure 5. Please check all table and figure.
  2. L387-389 and Table 3: There is a lack of statistical results for Kn. The authors are recommended to provide the relevant statistical analysis to support the interpretation of Kn data.
  3. Figure 4: Full explains of all abbreviations should be provided in figure legend.
  4. Figure 7: The Kaplan–Meier survival curve alone serves as a visual representation of survival rates and does not directly provide statistical significance for comparisons between groups. The log-rank method was used for comparisons in Kaplan–Meier survival curve. However, in Lines 252–254, the description of the Cox proportional hazards model and the method used for group comparisons is insufficient and should be clarified.

Results

  1. 5. Effect of time on growth performance and feed utilization of experimental diets & 3.6. Effects of experimental diets on hepatosomatic index: The T values are directly embedded within the text, making the results difficult to read and interpret. For instance, Lines 340–341 report t values for final weight and weight gain; however, only four t values are provided. Considering there are four groups and two parameters, there should be a total of eight statistical results. Please check the manuscript. Moreover, it is recommended to present the statistical results in a more organized format, such as a table.

Discussion

  1. The antibacterial mechanism of AND should be introduced and discussed.
  2. The authors state that sustained release is the key mechanism of NLCs; however, further clarification is needed. Does AND-NLC release mainly occur in the intestine or bloodstream? Are there other possible mechanisms? For instance, protect from digestive degradation, enhance intestinal absorption, and lymphatic transport avoid of first-pass metabolism should also be discussed.

Author Response

Reviewer 4

The present study was conducted to investigate the effects of andrographolide-loaded nanostructured lipid carriers on Nile tilapia, and some positive results have been observed. However, the manuscript should be further improved.

Introduction

Comment 1: The difference of FNA07, FPrA02, and ENC06, such as infection syndrome or primary genetic variation locations, should be introduced.

Response: We thank the reviewer for this valuable suggestion. In response, we have revised the Introduction section to include a brief explanation of the distinctions among the S. agalactiae strains FNA07, FPrA02, and ENC06. While all three belong to serotype Ia and sequence type 7 (ST7), they differ slightly in virulence traits, host-pathogen interactions, and geographic origins. These strains were selected to reflect genetic diversity and relevance to Thai aquaculture outbreaks. We have now included a sentence to highlight these differences for better contextual understanding.

In the revised manuscript:

Lines 76-80:

Although these strains share serotype Ia and sequence type ST7, they exhibit minor genetic variation in virulence loci such as scpB and cylE, which may influence their pathogenicity [14,15]. Experimental infections in Nile tilapia have demonstrated variable clinical outcomes among strains, supporting the notion that isolates like ENC06 may cause more severe systemic disease [16,17]. 

Method

Comment 2: L177: Please directly list the test substances. The title of Table 2 states that both AND and AND-NLC were tested; however, the table only presents results for NLC and AND-NLC, which is inconsistent. Please also verify the consistency and accuracy of the data presented in the manuscript (L299-305).

Response: Thank you for your valuable observation. We have revised the manuscript to directly list the tested substances as requested. Additionally, we acknowledge the typographical error in the first row of Table 2, where "NLC" was mistakenly written instead of "AND." This has now been corrected in the revised table. Therefore, the title of Table 2—"Inhibition zone diameter (IZD), minimum inhibitory concentration (MIC), and minimum bactericidal concentration (MBC) of Andrographolide (AND) and Andrographolide-loaded Nanostructured Lipid Carriers (AND-NLC) against S. agalactiae strains (FPrA02, FNA07, ENC06)"—along with the interpretation in lines 321–325, is now accurate and consistent with the presented data.

In the revised manuscript:

Lines 193-194: Wells with a diameter of 6 mm were created using sterile pipette tips, and 50 µL of either AND or AND-NLC was introduced into each well.

Lines 341-343:

Table 2. Inhibition zone diameter (IZD), minimum inhibitory concentration (MIC), and minimum bactericidal concentration (MBC) of Andrographolide (AND) and Andrographolide-loaded Nanostructured Lipid Carriers (AND-NLC) against S. agalactiae strains (FPrA02, FNA07, ENC06).

Formulation

IZD (mm)

MIC (ppm)

MBC (ppm)

FPrA02

FNA07

ENC06

FPrA02

FNA07

ENC06

FPrA02

FNA07

ENC06

AND

6±0a

6±0a

6±0a

1,000

1,000

1,000

2,000

2,000

2,000

AND-NLC

13±2b

13±2b

14±1b

>2,000

>2,000

>2,000

>2,000

>2,000

>2,000

Comment 3: 10. Growth performance and feed utilization: In the formula of PER, the unit of protein intake is gram not %.

Response: Thank you for your careful review. We agree with your observation. The unit of protein intake in the Protein Efficiency Ratio (PER) formula is indeed grams (g), not percentage (%). We have revised the text accordingly in the revised manuscript in equation 7 to reflect the correct unit for protein intake used in the PER calculation.

Comment 4: L234: The calculation of Wc is not described clearly. Is the regression used for Wc calculation based on all treatment fish at day 60, or derived from another reference dataset?

Response: Thank you for the comment. We appreciate your observation and would like to clarify that the weight–length regression equation used to calculate Wc was derived from all fish across treatment groups at day 60 of the experiment. The data were not obtained from any external or reference dataset. To enhance clarity, we have revised the description in the Materials and Methods section accordingly.

In the revised manuscript:

Lines 254-257: …. Wc is the expected weight calculated from the length–weight regression equation (W = aL^b). The regression was derived from all experimental fish at day 60 to represent the overall weight–length relationship.

Comment 5: 12. Disease resistance of Nile tilapia against Sagalactiace ENC06: Three strains, namely FNA07, FPrA02, and ENC06, were used for the antibacterial test; however, only ENC06 was used for the challenge experiment. Please clarify the rationale for selecting only ENC06 for the challenge.

Response: Thank you for your valuable comment. We have clarified the rationale in the revised manuscript. ENC06 was selected for the challenge trial based on its slightly higher inhibition zone in the antibacterial assay and its well-documented association with severe streptococcosis outbreaks in Thai Nile tilapia aquaculture. Among the tested strains, ENC06 exhibited a slightly larger inhibition zone (14 mm) than FNA07 and FPrA02 (13 mm each), indicating marginally higher susceptibility to the tested compound. Moreover, ENC06 is one of the most frequently isolated serotype Ia strains in Thailand and has been previously associated with higher virulence and systemic symptoms in experimental infections. This epidemiological and pathogenic relevance made ENC06 a practical and representative model for assessing the protective efficacy of the formulation. We have included this explanation at the beginning of Section 2.12 in the revised manuscript.

In the revised manuscript:

Lines 261-265: ENC06 was selected for the challenge experiment based on its marginally higher zone of inhibition in the antibacterial assay and its documented association with severe streptococcosis outbreaks in Thai Nile tilapia aquaculture [16,17]. This strain is widely recognized for its virulence and epidemiological relevance, making it a suitable model for evaluating disease resistance.

Table & Figure

Comment 6: Significance letters are incomplete in several figure, such as SGR and FCR of day 30 in Figure 3 and Figure 5. Please check all table and figure.

Response: Thank you for your careful observation. We have thoroughly reviewed all figures and tables in the revised manuscript. The significance letters in Figures 3 and 5, particularly for SGR and FCR on Day 30, have now been corrected to ensure completeness and clarity. All figures and tables have been cross-checked to confirm consistency in the use of superscript letters for statistical significance. These revisions are now accurately reflected in the updated manuscript.

Comment 7: L387-389 and Table 3: There is a lack of statistical results for Kn. The authors are recommended to provide the relevant statistical analysis to support the interpretation of Kn data.

Response: Thank you for the insightful comment. We have now included the relevant statistical analysis for Kn in Table 3, with appropriate superscripts to indicate significant differences between treatment groups. Additionally, we have updated the results section (Lines 421–426) to reflect these findings more accurately.

In the revised manuscript:

Lines 427-432: The Kn was significantly higher in the AND group (1.229), indicating superior body condition, followed by the AND-NLC group (1.053). The NLC group showed a significantly lower Kn value (0.971) compared to the other treatments. These results suggest that both AND and AND-NLC improve fish condition, with AND-NLC showing enhanced potential as a functional feed additive in tilapia aquaculture.

Comment 8: Figure 4: Full explains of all abbreviations should be provided in figure legend.

Response: Thank you for pointing this out. We have revised the figure legend for Figure 4 to include full definitions of all abbreviations for clarity and ease of understanding. The updated legend is as follows:

In the revised manuscript:

Figure 4. Pearson correlation heatmap showing relationships among nanoparticle characteristics, antibacterial activity, growth, feed utilization, physiological indices, and disease resistance parameters in Nile tilapia. Strong positive and negative correlations are indicated by color intensity and direction. Treatment (TRT); nanoparticle characteristics: particle size (Size), polydispersity index (PDI), and zeta potential (ZP); antibacterial activity: inhibition zone (IZ), minimum inhibitory concentration (MIC), and minimum bactericidal concentration (MBC); growth performance: final weight (FW), final length (FL), weight gain (WG), and specific growth rate (SGR); feed utilization: feed intake (FI), feed consumption ratio (FCR), protein efficiency ratio (PER); physiological indices: hepatosomatic index (HSI), length-weight relationship (LWR), and relative condition factor (Kn); and disease resistance: mortality (M), survival (S), and relative percent survival (RPS) in Nile tilapia.

Comment 9: Figure 7: The Kaplan–Meier survival curve alone serves as a visual representation of survival rates and does not directly provide statistical significance for comparisons between groups. The log-rank method was used for comparisons in Kaplan–Meier survival curve. However, in Lines 252–254, the description of the Cox proportional hazards model and the method used for group comparisons is insufficient and should be clarified.

Response: Thank you for your insightful comment. We confirm that both the Kaplan–Meier survival analysis and the Cox proportional hazards model were used in this study. The Log-Rank (Mantel–Cox) test was used to assess the statistical significance of differences in survival curves among treatment groups, while the Cox proportional hazards model was applied to estimate the hazard ratios and assess the relative risk of mortality across groups. The description in the statistical analysis section has now been revised to clearly reflect the use of both methods.

In the revised manuscript:

Lines 279-290:

2.13. Statistical analysis 

Statistical analyses were conducted using SPSS software (version 26; IBM Corp., Armonk, NY, USA). One-way analysis of variance (ANOVA) followed by Tukey’s Honest Significant Difference (HSD) test was used to determine significant differences among treatment groups for growth, feed utilization, and physiological parameters. Kaplan–Meier survival analysis was used to estimate cumulative survival, and Log-Rank (Mantel–Cox) tests were applied to compare survival curves between groups. Additionally, the Cox proportional hazards model was employed to assess the relative risk of mortality (hazard ratios) among different dietary treatments. The Student’s t-test was used to evaluate differences between two time points (30 and 60 days). Pearson’s correlation coefficients (r) were calculated to examine relationships among nanoparticle characteristics, antibacterial activity, growth performance, and disease resistance. A p-value less than 0.05 was considered statistically significant.

Results

Comment 10: 5. Effect of time on growth performance and feed utilization of experimental diets & 3.6. Effects of experimental diets on hepatosomatic index: The T values are directly embedded within the text, making the results difficult to read and interpret. For instance, Lines 340–341 report t values for final weight and weight gain; however, only four t values are provided. Considering there are four groups and two parameters, there should be a total of eight statistical results. Please check the manuscript. Moreover, it is recommended to present the statistical results in a more organized format, such as a table.

Response: Thank you for pointing this out. We acknowledge that embedding multiple t-values within the text may hinder readability. To improve clarity, we have reformatted the relevant sections and provided the complete t-statistics in a supplementary table (Supplementary Table S1). This table summarizes the paired t-test results for each treatment group at 30 and 60 days, across all growth and feed utilization parameters (FW, WG, SGR, FI, FCR, PER), including the hepatosomatic index (HSI).

Regarding the number of t-values reported: for FW and WG, the values were identical because both are directly correlated and derived from the same weight measurements. Nonetheless, we have now listed each value separately for completeness and transparency.

In the revised manuscript:

Lines 387-396: All treatment groups showed significant increases in final weight (FW) and weight gain (WG) from day 30 to day 60, indicating time-dependent improvement in growth. The specific growth rate (SGR) increased significantly over time only in the NLC group, while remaining unchanged in the other groups. Feed intake (FI) also increased significantly in all groups over time, reflecting elevated metabolic demands with growth progression. Notably, feed conversion ratio (FCR) was significantly reduced in the AND-NLC group at 60 days, suggesting enhanced feed efficiency, whereas other groups showed no significant change. Similarly, protein efficiency ratio (PER) improved significantly across all treatments, with the most pronounced increase observed in the AND-NLC group. The detailed t-test results for each growth and feed utilization parameter are provided in Supplementary Table S1.

Lines 401-407: At 30 days, no significant differences in HSI were observed among treatments. However, by day 60, the AND-NLC group exhibited significantly higher HSI values compared to all other treatments. Significant increases in HSI were also observed within the AND, AND-NLC, and NLC groups over time, indicating enhanced liver mass potentially associated with improved nutrient metabolism or liver function. The detailed paired t-test results for HSI across time points are provided in Supplementary Table S1.

Table S1. Paired t-test results comparing growth performance and feed utilization parameters (Final Weight [FW], Weight Gain [WG], Specific Growth Rate [SGR], Feed Conversion Ratio [FCR], and Protein Efficiency Ratio [PER]) of Nile tilapia between day 30 and day 60 across different dietary treatments (Control, andrographolide (AND), andrographolide loaded nanostructured lipid carriers (AND-NLC) and nanostructured lipid carriers (NLC)).

Parameters

t

df

p

FW

Control

-10.177

58

0.001

AND

-12.297

0.001

AND-NLC

-10.489

0.001

NLC

-9.378

0.001

WG

Control

-10.177

58

0.001

AND

-12.297

0.001

AND-NLC

-10.489

0.001

NLC

-9.378

0.001

SGR

Control

1.472

58

0.073

AND

1.526

0.066

AND-NLC

-0.421

0.338

NLC

2.621

0.006

FCR

Control

0.576

58

0.284

AND

1.778

0.040

AND-NLC

3.265

0.001

NLC

-0.156

0.438

PER

Control

-10.177

58

0.001

AND

-12.297

0.001

AND-NLC

-10.489

0.001

NLC

-9.378

0.001

PER

Control

-0.845

15

0.206

AND

-6.043

0.001

AND-NLC

-2.175

0.023

NLC

-2.770

0.007

Discussion

Comment 11: The antibacterial mechanism of AND should be introduced and discussed.

Response: Thank you for your valuable comment. We agree with your suggestion and have revised the discussion to include a detailed explanation of the antibacterial mechanism of AND. We have incorporated relevant literature to clarify its mode of action, particularly focusing on its interaction with bacterial membranes, inhibition of quorum sensing, and disruption of vital cellular processes. These mechanisms are further supported by recent studies that highlight the molecular targets and bactericidal properties of diterpenoids like AND.

In the revised manuscript:

Lines 521-528: AND, a labdane diterpenoid, exerts its antibacterial effect through multiple mechanisms. One of the primary modes of action involves disruption of bacterial membrane integrity, leading to leakage of intracellular contents and cell lysis [57]. AND has also been shown to inhibit the bacterial quorum sensing (QS) system, which plays a crucial role in virulence regulation, biofilm formation, and antibiotic resistance [58]. Furthermore, AND interferes with key bacterial metabolic pathways and inhibits DNA and protein synthesis [59]. These multimodal actions not only confer broad-spectrum antibacterial activity but also reduce the likelihood of resistance development.

Comment 12: The authors state that sustained release is the key mechanism of NLCs; however, further clarification is needed. Does AND-NLC release mainly occur in the intestine or bloodstream? Are there other possible mechanisms? For instance, protect from digestive degradation, enhance intestinal absorption, and lymphatic transport avoid of first-pass metabolism should also be discussed.

Response: We thank the reviewer for this insightful comment. We agree that further clarification regarding the release site and additional mechanisms contributing to the enhanced bioavailability of AND-NLC is necessary. In the revised manuscript, we have expanded the discussion to explain that AND-NLC is primarily designed for intestinal release and absorption, where it benefits from protection against digestive degradation, improved epithelial permeability, and potential lymphatic transport that helps avoid first-pass hepatic metabolism. These mechanisms collectively contribute to the enhanced systemic availability, metabolic efficiency, and biological effects of AND in fish.

In the revised manuscript:

Lines 545-559: Beyond the sustained release, several complementary mechanisms contribute to the enhanced bioavailability and biological activity of AND-NLC in fish. Upon ingestion, NLCs are expected to release AND primarily in the intestinal tract, where the lipid matrix of the nanocarrier provides protection against acidic pH and enzymatic degradation in the stomach, ensuring a greater amount of intact compound reaches the absorption site [72]. The small particle size and low PDI enhance interaction with intestinal microvilli and promote uptake via transcellular and paracellular pathways [73]. Additionally, NLCs have been shown to improve intestinal permeability and epithelial absorption of lipophilic compounds, facilitating systemic circulation [74]. Some studies in fish suggest that lipid-based nanoparticles may also promote lymphatic transport, helping to bypass first-pass hepatic metabolism and increasing systemic bioavailability [75]. These mechanisms jointly support the delayed but significant improvements in growth and feed efficiency observed in the AND-NLC group after 60 days. Similar advantages have been reported with nano-curcumin [73], nano-omega-3 fatty acids [74], and Laurus nobilis essential oil nano-particles [76] in Tilapia sp.

Round 2

Reviewer 1 Report

Comments and Suggestions for Authors

Thanks to the authors for their careful revision. Although I am still not very satisfied with the experimental design of this paper, the authors did solve most of my concerns.